# Active Flutter Suppression of Smart-Skin Antenna Structures with Piezoelectric Sensors and Actuators

Chang-Yull Lee [1] and Ji-Hwan Kim [2,*]

[1] Department of Aerospace Engineering, Inha University, Incheon 22212, Korea; cylee@inha.ac.kr
[2] Institute of Advanced Aerospace Technology, Department of Aerospace Engineering,
    Seoul National University, Seoul 08826, Korea
* Correspondence: jwhkim@snu.ac.kr; Tel.: +82-2-880-7383

**Abstract:** A smart-skin antenna structure is investigated for active flutter control with piezoelectric sensors and actuators. The skin antenna is designed as a multilayer sandwich structure with a dielectric polymer to perform the role of antenna or radar structures. The governing equations are developed according to the first-order shear deformation theory, and von Karman strain–displacement relationships are used for the moderate geometrical nonlinearity. To consider the supersonic airflow, first-order piston theory is performed for the aerodynamic pressures. The linear quadratic regulator (LQR) method is applied as a control algorithm, and Newmark's method is studied to obtain the numerical results. In the present study, the effects of placements and shape of piezoelectric patches are discussed on the flutter control of the model in detail. In addition, the numerical results show that the skin antenna model can effectively suppress the panel flutter behaviors of the model, optimal conditions of piezoelectric patches are obtained for skin antenna structures.

**Keywords:** smart-skin antenna structure; active flutter control; LQR control algorithm; piezoelectric sensor and actuator

## 1. Introduction

A stealth technology for military aircrafts has been one of the hot topics, and advanced technologies for the new composition have been suggested over the last several decades. Especially, multifunctional aircraft structures (MAS) have been developed by many engineers. Further, integrating airframe structures with functional applications lead to reduction of the weight, radar cross section (RCS) and volume. Additionally, the smart-skin antenna structure as a kind of MAS has been widely investigated to improve both structural efficiency and antenna performance.

In this regards, Varadan and Varadan [1] suggested the smart-skin antenna structure as a conformable load-bearing antenna structures (CLAS) using composite materials. Because the development of stealth technology for military aircrafts is very important for the safety of the countries, many researchers have tried to develop and studied various types of antenna structures. Yao et al. [2] made the 3D integrated microstrip antenna model, and investigated its radiation pattern. Jeon et al. [3] made another model and tested the buckling behaviors of the structure experimentally. Additionally, Yoon et al. [4] proposed and studied a CLAS, and performed experimental and numerical results for various cases. Additionally, Daliri et al. [5] investigated composite materials for the mechanical and electromagnetic performance of slot log-spiral antenna structures. Lee and Kim studied the thermal stability regions and obtained the limit cycle oscillation behaviors of smart skin antenna structures [6]. Further, Yoo and Kim [7] investigated optimal conditions of models for thermal buckling and vibration. Up to now, numerous works have studied the dynamic responses of structures under aerodynamic loads. Panel flutter

is the dynamic instability of the thin panel of flight vehicles with inertia force, elastic force and aerodynamic pressure [8–10].

Antenna systems embedded in an airframe have many kinds of advantages. However, integrated antenna models can vibrate due to aerodynamic loads. Then, the models will have severe deviations of the signal information. It may also result in the degradation of the antenna's performance [11]. Recently, some engineers have pointed out the necessity of understanding the control method of the structures [12].

Piezoelectrics are the most popular materials, and piezoelectric devices have been actively studied as actuators/sensors. Lam et al. [13] studied the vibration control of a composite plate with distributed piezoelectric sensors and actuators. Lee et al. [14] presented active vibration suppression of stiffened composite panels using piezoelectric sensors and actuators. Liu et al. [15] investigated the dynamic response of laminated composite plates with piezoelectric materials subjected to mechanical and electrical loadings. Recently, research on vibration control for space application has been actively performed using smart materials [16–18].

Though numerous papers have investigated wide aspects of skin antenna structures have been investigated widely up to now, the active flutter controls of antennas have been studied in a limited range. In this work, a smart-skin antenna structure is studied for panel flutter suppression using piezoelectric sensors and actuators. The position of the piezoelectric patch is significant for effective vibration control. For the antenna structures, the most important thing is that the other devices, such as sensors/actuators, must not block the antenna's functional elements for dealing with electromagnetic radiation. In this regard, we focused on the positions of the patches and discuss various studies on them in detail. Primarily, the model is investigated to determine the optimal positions of piezoelectric patches with reliable performance for panel flutter suppression. Additionally, the panel flutter suppressions of the designed structure are investigated. Additionally, aerodynamic pressure is obtained by the first-order piston theory in the supersonic regions. Additionally, linear quadratic regulator (LQR) control algorithm is chosen, and Newmark's method is applied to obtain the numerical results. In a conventional antenna, the electromagnetic parameters of the antenna model are very important. However, when the aircraft fuselage acts as an antenna, the structural deformations of the body become significant factors in the performance of the antenna, as reported by numerous research papers [11,12]. Therefore, the structural behaviors of the smart skin antenna, not the electromagnetic characteristics of the antenna, are our focus in this study.

## 2. Formulations

A structure is proposed with basic concepts that can be expected to be required for the antenna function, as detailed in in Ref. [19]. Figure 1 shows a smart-skin antenna model with five whole layers.

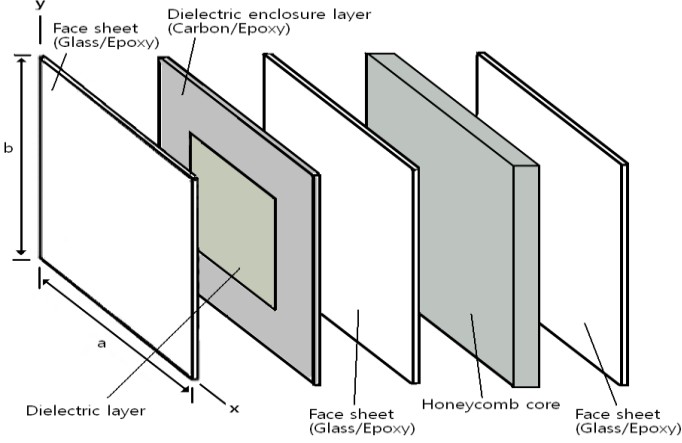

**Figure 1.** The model of smart-skin antenna structure.

The model consists of five layers. In addition, the width, length and thickness are $b$, $a$ and $h$, respectively. From the left part of the model, the staking sequences are the face sheet, the dielectric layer with a dielectric enclosure, a face sheet, the honeycomb core and another face sheet. The face sheets can protect the dielectric components from various external loads. This model is an asymmetric multi-layered structure. Therefore, the total displacement of the model can be calculated as the summation of the displacement for shear deformation of the core and the displacement due to bending of the model. Additionally, the honeycomb core transmits shear between the sheets, and conducts it as an air gap.

*2.1. Constitutive Equations*

The first-order shear deformation theory (FSDT) and the von Karman strain–displacement relations are applied.

$$
\begin{aligned}
\mathbf{e} &= \boldsymbol{\varepsilon}^0 + z\boldsymbol{\kappa} \\
&= \left\{ u_0,_x \quad v_0,_y \quad u_0,_y + v_0,_x \right\} + \frac{1}{2}\left\{ w_0,_x^{\,2} \quad w_0,_y^{\,2} \quad 2w_0,_x\, w_0,_y \right\} \\
&\quad + z\left\{ \phi_x,_x \quad \phi_y,_y \quad \phi_x,_y + \phi_y,_x \right\} \\
\boldsymbol{\gamma}^T &= \{\gamma_{yz} \quad \gamma_{xz}\}^T = \{ w_0,_y + \phi_y \quad w_0,_x + \phi_x \}^T
\end{aligned}
\tag{1}
$$

where $u$, $v$ and $w$ are the midplane displacements, respectively. In addition, $\phi_x$ and $\phi_y$ are the perpendicular to the longitudinal plane. Additionally, $\boldsymbol{\varepsilon}^0$, $\boldsymbol{\kappa}$ and $\boldsymbol{\gamma}$ are the inplane strain vectors at the midplane, the curvature strain vectors, and transverse shear strain vectors, respectively.

The stress of the $k^{th}$ layer can be written by the transformation of coordinates as follows:

$$
\{\sigma\} = [Q]\big(\{\varepsilon\} - \Delta T\{\alpha\} - e\{d\}\big)
\tag{2}
$$

where $Q$, $\alpha$, $e$ and $d$ are transformed reduced lamina stiffness matrix, thermal expansion coefficient, electric field and electromechanical coefficient, respectively.

The constitutive equation for laminate plates with thermal and piezoelectric conditions can be derived as follows [20]:

$$
\begin{Bmatrix} \mathbf{N_b} \\ \mathbf{M_b} \end{Bmatrix} = \begin{bmatrix} \mathbf{A} & \mathbf{B} \\ \mathbf{B} & \mathbf{D} \end{bmatrix} \begin{Bmatrix} \boldsymbol{\varepsilon}^0 \\ \boldsymbol{\kappa} \end{Bmatrix} - \begin{Bmatrix} \mathbf{N_{\Delta T}} \\ \mathbf{M_{\Delta T}} \end{Bmatrix} - \begin{Bmatrix} \mathbf{N_{\Delta P}} \\ \mathbf{M_{\Delta P}} \end{Bmatrix}
$$
$$
\mathbf{Q} = \mathbf{A_s}\boldsymbol{\gamma}
\tag{3}
$$

where $\mathbf{N_b}$, $\mathbf{M_b}$ and $\mathbf{Q}$ mean the inplane force, moment and transverse shear force resultant vectors, respectively. Meanwhile, $(\mathbf{N_{\Delta T}}, \mathbf{M_{\Delta T}})$ and $(\mathbf{N_{\Delta P}}, \mathbf{M_{\Delta P}})$ are defined as the temperature and piezoelectric dependent quantities as follows,

$$
\begin{aligned}
(\mathbf{N_{\Delta T}}, \mathbf{M_{\Delta T}}) &= \sum_{k=1}^{n} \int_{z_{k-1}}^{z_k} \bar{\mathbf{Q}}_k \alpha_k \Delta T (1, z)\, dz \\
(\mathbf{N_{\Delta P}}, \mathbf{M_{\Delta P}}) &= \sum_{k=1}^{n} \int_{z_{k-1}}^{z_k} \bar{\mathbf{Q}}_k d_k e_k (1, z)\, dz
\end{aligned}
\tag{4}
$$

In addition, $\mathbf{A}$, $\mathbf{B}$, $\mathbf{D}$ and $\mathbf{A_s}$ denote the extensional matrix, bending-extension coupling matrix, bending and shear stiffness matrices, respectively. Further, $\bar{\mathbf{Q}}_k$, $\bar{\mathbf{d}}_k$ and $E_{1k}$ are the transformed reduced stiffness matrix, the transformed piezoelectric constant vector and the electric field, respectively.

$$(\mathbf{A}, \mathbf{B}, \mathbf{D}) = \sum_{k=1}^{n} \int_{z_{k-1}}^{z_k} \bar{\mathbf{Q}}_k \left(1, z, z^2\right) dz,$$

$$\mathbf{A_s} = \sum_{k=1}^{n} \kappa_p \int_{z_{k-1}}^{z_k} \bar{\mathbf{Q}}_k \, dz \tag{5}$$

where $\kappa_p$ stands for shear correction factor.

### 2.2. Equations with Aerodynamic Flows

The equations of motion for the panel flutter analysis are derived using the principle of virtual work as

$$\delta W = \delta W_{\text{int}} - \delta W_{ext} = 0 \tag{6}$$

where $\delta W_{ext}$ and $\delta W_{\text{int}}$ are the external and internal virtual works, respectively.

Virtual works can be rewritten in terms of structural parts and external loads.

$$\delta W_{int} - \delta W_{ext} = \int_V \{\delta e\}^T \{\sigma\} dV - \left[-\{\delta d\}^T [M]\{\ddot{d}\} + \{\delta d\}^T \{f\}\right] = 0 \tag{7}$$

where $\{\sigma\} = \{\sigma_{xx}, \sigma_{yy}, \tau_{xy}\}^T$ and $\{d\} = \{u, v, w, \phi_x, \phi_y\}^T$ denote the stress and displacement vectors. While $[M]$ and $\{f\}$ represent the mass matrix and external force vector, respectively.

Then, the internal virtual work,

$$\begin{aligned}
\delta W_{int} &= \int_V \{\delta e\}^T \{\sigma\} dV \\
&= \int_A \left(\{\delta\varepsilon_m\}^T \{N_b\} + \{\delta\kappa\}^T \{M_b\} + \{\delta\gamma\}^T \{Q\}\right) dA \\
&= \{\delta d\}^T \left([K] - [K_{\Delta T}] + \tfrac{1}{2}[N1] + \tfrac{1}{3}[N2]\right)\{d\} - \{\delta d\}^T \{P_{\Delta T}\}
\end{aligned} \tag{8}$$

where $\{\varepsilon_m\} = \{\varepsilon_{xx}, \varepsilon_{yy}, \gamma_{xy}\}^T$ denotes the strain, while $[K_{\Delta T}]$, $[K]$, $[N1]$, $[N2]$ and $\{P_{\Delta T}\}$ represent the thermal geometric stiffness matrix, the linear elastic stiffness matrix, the first-order nonlinear stiffness matrix, the second order nonlinear stiffness matrix and thermal load vectors, respectively.

Then, external virtual work is

$$\begin{aligned}
\delta W_{ext} &= -\{\delta d\}^T [M]\{\ddot{d}\} + \{\delta d\}^T \{f\} \\
&= \int_A [-I_0 \left(\ddot{u}_0 \delta u_0 + \ddot{v}_0 \delta v_0 + \ddot{w}_0 \delta w_0\right) \\
&\quad - I_1 \left(\ddot{u}_0 \delta\phi_x + \ddot{\phi}_x \delta u_0 + \ddot{v}_0 \delta\phi_y + \ddot{\phi}_y \delta v_0\right) \\
&\quad - I_2 (\ddot{\phi}_x \delta\phi_x + \ddot{\phi}_y \delta\phi_y) + p_a \delta w] dA
\end{aligned} \tag{9}$$

where, $(I_0, I_1, I_2) = \int_{-h/2}^{h/2} \rho(1, z, z^2) dz$ and $p_a$ are the moment of inertias and aerodynamic pressure, respectively. Further, $p_a$ stands for the aerodynamic force for a supersonic air flow. It is valid for $\sqrt{2} < M_\infty < 5$ according to the first-order piston theory, as in Ref. [21]. Additionally, it can be expressed as $p_a(x, y, t) = -\frac{\rho_a V_\infty^2}{\sqrt{M_\infty^2 - 1}} \left\{\frac{\partial w}{\partial x} + \left(\frac{M_\infty^2 - 2}{M_\infty^2 - 1}\right) \frac{1}{V_\infty} \frac{\partial w}{\partial t}\right\}$. Here, $V_\infty$, $\rho_a$ and $M_\infty$ are the air density, the air flow speed and Mach number, respectively.

Furthermore, aerodynamic pressure loads acting on the panel are

$$\{\delta d\}^T \{f\} = \int_A p_a \delta w \, dA$$

$$= -\int_A \left( \lambda \frac{D}{a^3} \frac{\partial w}{\partial x} + \frac{g_a}{\omega_0} \frac{D}{a^4} \frac{\partial w}{\partial t} \right) \delta w \, dA = -\{\delta d\}^T \left( \lambda \left[ A_f \right] \{d\} + \frac{g_a}{\omega_0} \left[ A_d \right] \{\dot{d}\} \right) \quad (10)$$

where $\lambda = \dfrac{\rho_a V_\infty^2 a^3}{\beta D_m}$ stands for the non-dimensional aerodynamic force. Additionally, $g_a$, $\omega_0$, $\left[ A_d \right]$ and $\left[ A_f \right]$ are the nondimensional aerodynamic damping component, convenient reference frequency, the damping matrix and the aerodynamic influence matrix, respectively [22].

### 2.3. Piezoelectric Sensors and Actuators

The piezoelectric coupling of $k^{th}$ layer in the electric and the elastic fields can be derived as the direct piezoelectric equation and the converse piezoelectric equation. Then, the components of a piezoelectric materials are obtained as follows [13]:

$$\boldsymbol{\sigma}_k = \mathbf{Q}_k \boldsymbol{\varepsilon}_k - \mathbf{e}_k^T \mathbf{E}_k$$
$$\mathbf{D}_k = \mathbf{e}_k \boldsymbol{\varepsilon}_k + \in_k \mathbf{E}_k \quad (11)$$

where $\mathbf{D}$, $\boldsymbol{\varepsilon}$, $\boldsymbol{\sigma}$ and $\mathbf{E}$ are electric displacement, strain, stress and electric field vectors. In addition, $\mathbf{Q}$, $\in$ and $\mathbf{e}$ are the elastic matrix, permittivity coefficients and piezoelectric constants, respectively.

Then, the electric potential $V$ is related to the electric field vector $\mathbf{E} = -\nabla V$. The voltage applied to the actuators. Then, the electric field vector $\mathbf{E}$ is obtained as follows:

$$\mathbf{E} = \begin{bmatrix} 0 & 0 & 1/h_a \end{bmatrix}^T V_a \quad (12)$$

where $V_a$ and $h_a$ are the applied with a voltage and the thickness of the actuator layer, respectively. The electric displacement $D_z$ can be obtained as follows:

$$D_z = e_{31} \boldsymbol{\varepsilon} \quad (13)$$

where $e_{31}$ stands for the piezoelectric constant, and the charge $q(t)$ activated on the sensor surface is the sum of the charges as follows:

$$q(t) = \int_S D_z dS \quad (14)$$

where $S$ stands for the surface area, and the sensor voltage output $V_S$ can be written as follows:

$$V_S(t) = G_c i(t) \quad (15)$$

where $G_c$ is the gain component. Then, the current $i(t)$ is the time derivative of the charge as flows:

$$i(t) = \frac{dq(t)}{dt} \quad (16)$$

where $q(t)$ is a total charge, as shown in Equation (14).

The piezoelectric patches can then be placed, either through being surface bonded or embedded into the substrate.

### 2.4. Governing Equation

Using Sections 2 and 3, the energy principle is applied for a model including the piezoelectric resultants as follows:

$$\begin{bmatrix} \mathbf{M}_{uu} & 0 \\ 0 & 0 \end{bmatrix}\begin{Bmatrix} \{\ddot{u}\} \\ \phi \end{Bmatrix} + \begin{bmatrix} \mathbf{K}_{uu} & \mathbf{K}_{u\phi} \\ \mathbf{K}_{\phi u} & \mathbf{K}_{\phi\phi} \end{bmatrix}\begin{Bmatrix} \{u\} \\ \phi \end{Bmatrix} = \begin{Bmatrix} \mathbf{F} \\ 0 \end{Bmatrix} \tag{17}$$

where $\mathbf{K}_{\phi\phi}$ and $\mathbf{K}_{uu}$ are the electric stiffness matrix and the elastic matrix, respectively. Further, $\mathbf{K}_{\phi u}$ and $\mathbf{K}_{u\phi}$ stand for the coupling matrices. Then, actuator and sensor equations can be derived as follows:

$$\begin{aligned} \{u\} &= \mathbf{K}_{uu}^{-1}\left(\mathbf{F} - \mathbf{K}_{u\phi}\phi_A\right) \\ \phi_s &= -\mathbf{K}_{\phi\phi}^{-1}\mathbf{K}_{\phi u}\{u\} \end{aligned} \tag{18}$$

where $\{\phi_s\}$ and $\{\phi_A\}$ mean electric displacement vectors of sensing and actuation, respectively. Finally, assembling the element equations provides the global dynamic governing equation as

$$\mathbf{M}_{uu}\{\ddot{u}\} + \mathbf{C}\{\dot{u}\} + \left(\mathbf{K}_{uu} - \mathbf{K}_{u\phi}\mathbf{K}_{\phi\phi}\mathbf{K}_{\phi u}\right)\{u\} = \mathbf{F} + \mathbf{K}_{u\phi}\phi_A \tag{19}$$

Then, the linear quadratic regulator (LQR) method is applied. The control gain can be derived as follows:

$$J = \frac{1}{2}\int_0^\infty \left(\boldsymbol{\xi}^T\mathbf{Q}\boldsymbol{\xi} + \phi_a^T\mathbf{R}\phi_a\right)dt \tag{20}$$

where $\mathbf{R}$ is the positive definite weighting matrix and $\mathbf{Q}$ is the semi-positive definite, respectively. The voltage can be derived as follows:

$$V_a(t) = -\mathbf{G}_c\boldsymbol{\xi} = -\mathbf{R}^{-1}\mathbf{B}_{st}^T\mathbf{P}\boldsymbol{\xi} \tag{21}$$

## 3. Numerical Results and Discussions

Numerical results are obtained for the positions and sizes of the piezoelectric patches for active flutter control of the smart-skin antenna structure. In this work, the skin is modeled as a multi-layered structure, as in Ref. [4]. The material properties are summarized in Table 1. Additionally, two additional piezoelectric layers (PZT G1195N) are attached to each of the layers, and the material properties of PZT are presented in Table 2. The uniform temperature is considered, and the reference temperature $T_0$ is 300 K. Furthermore, we considered clamped panels with a thickness ratio ($a/h$) as 100. The thickness ratio of the piezoelectric layers ($hp/h$) was chosen as 0.1 for all cases and the control values were $Q = 10$ and $R = 1$ unless a comment is made to indicate otherwise.

### 3.1. Code Verifications

To verify the simulations, three cases were performed. Firstly, Figure 2 presents the plots of natural frequencies for two ply angles (0° and 45°) according to the relative sizes of the dielectric part (dielectric region/total area of dielectric layer). Then, the natural frequencies decrease as the dielectric portions increase, and the numerical results show good agreements with the data in Ref. [7].

**Table 1.** Material properties of the smart-skin antenna structure [4].

|  | G/E | C/E | Phenol | Honeycomb |
|---|---|---|---|---|
| $E_1$ | 24 Gpa | 67 Gpa | 7.2 Gpa | 0.09 Mpa |
| $E_2$ | 28 Gpa | 57 Gpa | 7.2 Gpa | 0.08 Mpa |
| $\nu_{12}$ | 0.105 | 0.103 | 0.3 | 0.3 |
| $G_{12}$ | 4.54 Gpa | 5.9 Gpa | 2.77 Gpa | 0.1 Mpa |
| $G_{13}$ | 1.0 Gpa | 1.0 Gpa | 2.77 Gpa | 19.7 Mpa |
| $G_{23}$ | 1.0 Gpa | 1.0 Gpa | 2.77 Gpa | 11.5 Mpa |

| | | | | |
|---|---|---|---|---|
| $\alpha_1$ | 9.7$^{-6}$/°C | 2.1$^{-6}$/°C | 75$^{-6}$/°C | 1.5$^{-6}$/°C |
| $\alpha_2$ | 17.7$^{-6}$/°C | 2.1$^{-6}$/°C | 75$^{-6}$/°C | 1.5$^{-6}$/°C |
| $\varrho$ | 2200 kg/m³ | 1450 kg/m³ | 9000 kg/m³ | 96.1 kg/m³ |

**Table 2.** Material properties of PZT G1195N piezoceramics [13].

| Properties | PZT Piezoceramic | T300/976 |
|---|---|---|
| Young's modulus (GPa):  $E_{11}$ | 63.0 | 150 |
| $E_{22} = E_{33}$ | 63.0 | 9.0 |
| Poisson's ratio:  $\nu_{12} = \nu_{13}$ | 0.3 | 0.3 |
| $\nu_{23}$ | 0.3 | 0.3 |
| Shear modulus (GPa):  $G_{12} = G_{13}$ | 24.2 | 7.10 |
| $G_{23}$ | 24.2 | 2.50 |
| Density (kg/m³):  $\rho$ | 7600 | 1600 |
| Piezoelectric constants (m/V):  $d_{31} = d_{32}$ | $254 \times 10^{-12}$ | - |
| Electrical permittivity (F/m):  $\in_{11} = \in_{22}$ | $15.3 \times 10^{-9}$ | - |
| $\in_{33}$ | $15.0 \times 10^{-9}$ | - |

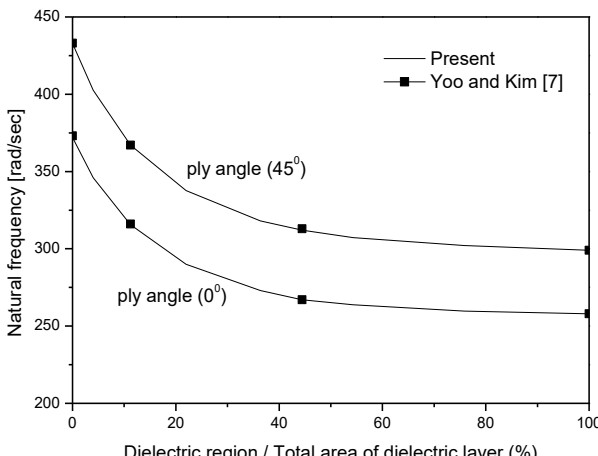

**Figure 2.** Natural frequencies for smart-skin antenna structure.

Next, Figure 3 shows the limit-cycle oscillation (LCO) amplitudes of the model to verify the time integration process. Newmark's method is used as 0.1 *ms* for the time step. Additionally, the deflection is the transverse deformation at *x/a = 3/4* and *y/b = 1/2* for maximum magnitude of the LCO. The numerical simulations show good agreement with the numerical data [21].

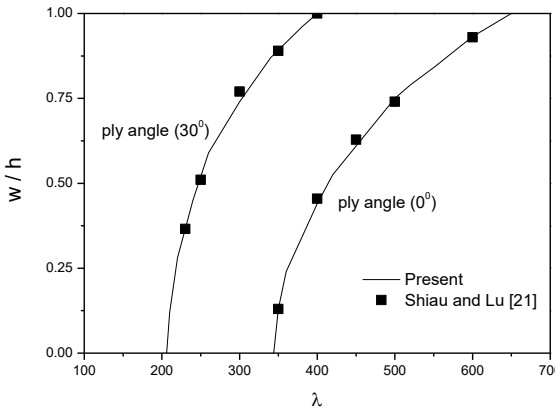

**Figure 3.** Amplitudes of limit cycle oscillation (LCO) of a composite model.

Finally, Figure 4 depicts the deflection behaviors of a cantilevered plate. The linear static analysis of the model with the upper layer and lower surface with piezoelectric materials placed on them. The input voltage is chosen as $10V$, and the stacking sequence of the model is [−45/45/−45/45]. The material properties for graphite/epoxy and PZT used in this case are presented in Table 2. The numerical result also shows good agreement with the data in Ref. [13].

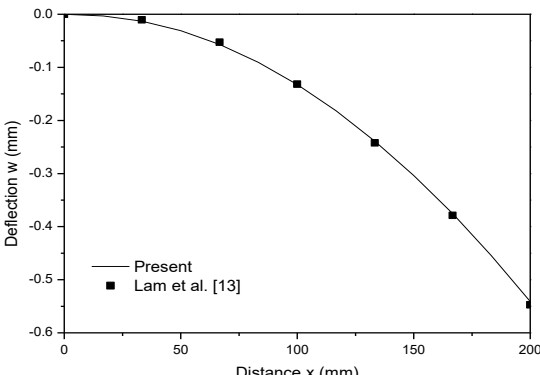

**Figure 4.** Deflection plot of the model with piezoelectric loads.

### 3.2. Positions and Sizes of the Patches

Generally, the dielectric portion as an antenna function is located at the center of the smart-skin antenna structure [23]. The dielectric layer is an essential part of the smart skin with various shapes of the dielectric region such as a circular [24], square [25] and rectangular [26] shapes. Thus, the pairs of piezoelectric sensors/actuators are not located in such a manner as to block the antenna elements, and the pairs should be used in patch-types instead of the layers. Prior to investigating the skin antenna structure, the honeycomb core model was studied with the distributed patches in order to obtain the effective control positions. Along with the results in Figures 5–8, we thereby determined the optimal conditions of each of the pairs of sensors/actuators for control. First of all, in order to easily understand the effects of various parameters in each restricted situation, only the vibration characteristics were considered without considering the aerodynamic nonlinearity. After then, the suppression of the flutter behaviors was studied using control method considering aerodynamic nonlinearity [27]. Figures 5–7 demonstrate the effectiveness of the vibration suppressions of the sandwich honeycomb core model. Due to the numerical

results in the early stage of vibration, they showed a limit cycle of motion with little change in amplitude without control. In Figures 5–7, the data show the effects with various parameters after the start of the control of the vibration motion. Figure 5 represents the effect of position for the patches on the responses of the model. As shown in Figure 5a, the patches are located in the center and corner positions of both the top and bottom surfaces of the panel, respectively. Figure 5b clearly shows that the vibration suppression efficiency is better as the piezoelectric materials are placed on the center portion of the structure. In other words, the performances of the patches are more efficient when they are closer to the center of the model.

To investigate the relationship between the sizes of the patches and the control effects, Figure 6 shows the vibration behaviors of the model with various sizes of actuator and sensor pairs, with $2 \times 2$ (6.25%) patches and $8 \times 8$ (100%) full meshes surface. As expected, the result presents that the vibration amplitude is more quickly reduced as the increase of the piezoelectric material sizes. In other words, the portions of piezoelectric parts on the structures are increasing, the control effect is enhanced. However, the quantity of the patch parts is limited in practical point of view, therefore optimal design for the location of the patches is important for active control.

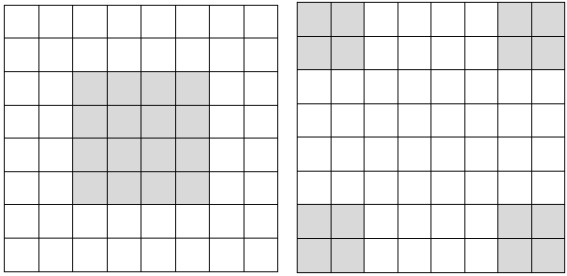

Model (Center) Model (Corner)

(**a**) Model (Center) and Model (Corner)

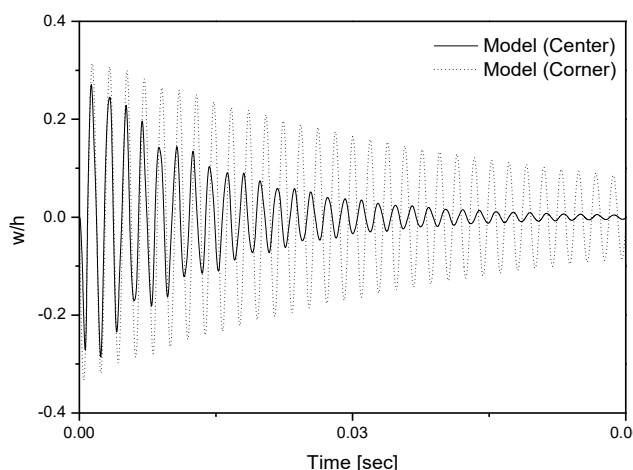

(**b**) Non-dimensional deflection with various locations for sensor/actuator pairs

**Figure 5.** Non-dimensional deflections according to various locations for sensor/actuator pairs.

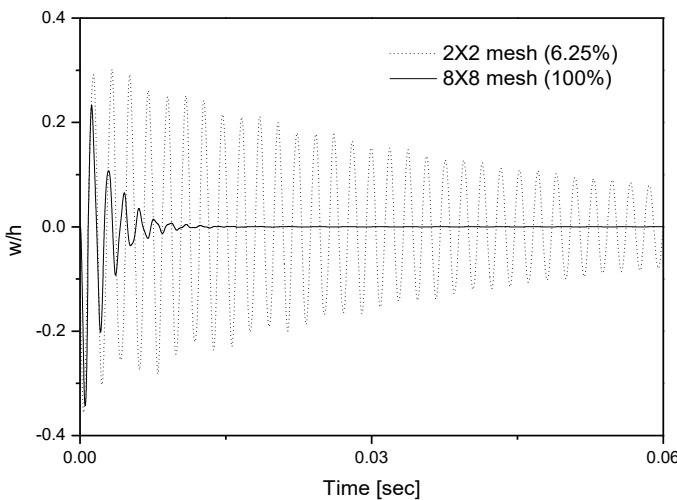

**Figure 6.** Non-dimensional deflections according to various sizes for the sensor/actuator pairs.

Figure 7 shows the effect of positions for piezoelectric materials for the inner layer and outer layer. In the two cases, the piezolayers are placed at the nearest and farthest points from the midplane, respectively. As the layers move farther from the midplane, the more quickly they are suppressed. This is due to the maximum moment generated by the layer, which occurs when the layers are placed farthest from the midplane. That is, the surface-bonded case presents more efficient control because the largest moment arm can be obtained with respect to the midplane.

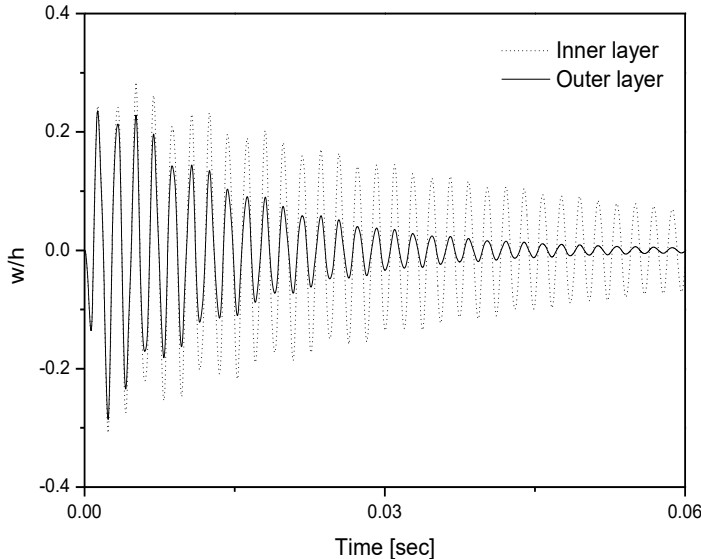

**Figure 7.** Non-dimensional deflections according to positions for sensor/actuator pairs through the thickness.

Now, the flutter suppressions of the structure are studied under supersonic flow. Generally, the typical panel LCO shape is different when exposed to the motion caused by vibration. Due to the aerodynamic flow, the peak point of deflections is moved backwards, and then the maximum panel deflection occurs at the point of *3/4* of the length

[21,22]. In this regard, Figure 8a shows that the patches are placed at the center, as shown in Model (I). However, the patches are placed at the *3/4* point, as shown in Model (II). Figure 8b presents the stable regions of the panels with various placements of the patches. Upon comparing the areas of stable regions of the two types, we find that the stability conditions of the Model (II) are better than those of Model (I) because the patch of Model (II) is placed near the peak amplitude for the model; thus, the performance of the patch is more effective, and similar results are presented in Ref. [23].

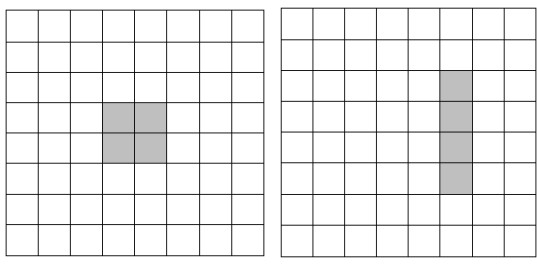

Model (I) Model (II)

(**a**) Model (I) and Model (II)

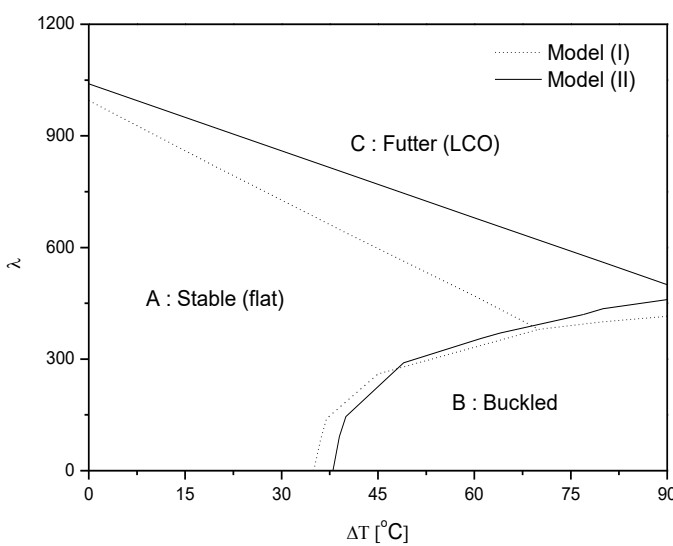

(**b**) Stability regions with Model (I) and (II)

**Figure 8.** Stability regions according to the positions sensor/actuator pairs.

### 3.3. Active Flutter Suppressions

Figure 9 shows the skin antenna models with piezoelectric sensor and actuator patches in supersonic airflow. In addition, three models were presented according to the size of the dielectric layer and the location of the patches. The center area indicates the dielectric antenna part, and the piezoelectric patches are located next to the part. To have the greatest possible effect on flutter control, the patches are placed at top and bottom layers and at *3/4* position from the airflow. As previously stated, the quantity of smart materials is limited in real time; thus, the patches are chosen as same area. In this regard, Figure 10a,b show flutter suppression via thermal and aerodynamic effects, respectively. Active control is started after 0.05 sec to compare the controlled responses with the uncontrolled responses.

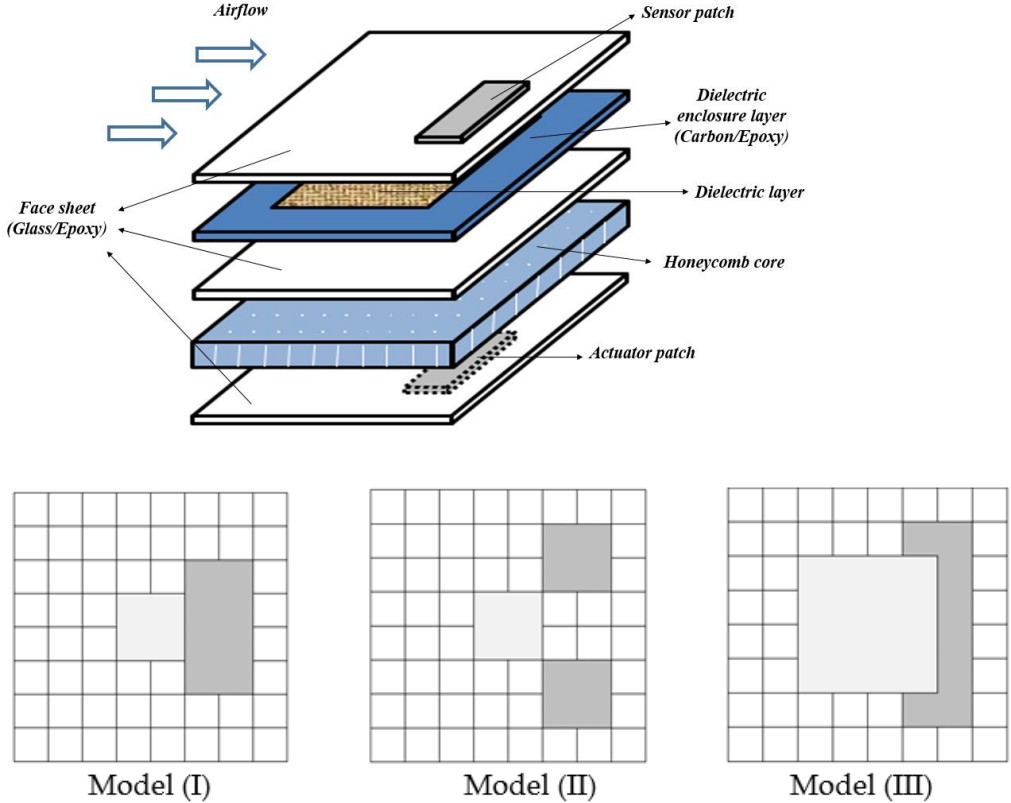

**Figure 9.** Three cases with various dielectric portion and piezoelectric patches.

Firstly, the thermal variation ( $\Delta T$ ) is selected as 0 and 30, and the aerodynamic load is chosen as $\lambda = 1200$ in Figure 10a. The result shows that the flutter motion of the model can be suppressed faster in low temperature conditions. On the other hand, the pressures ( $\lambda$ ) in Figure 10b are chosen as 1200 and 1600 without considering thermal effect. In this case, flutter suppression is more difficult with high pressure loads. It can be seen that the oscillation of the structure is reduced more quickly with low temperature and aerodynamic pressure. Additionally, the results of the bold lines are dealt with in the same conditions as in both Figure 10a,b.

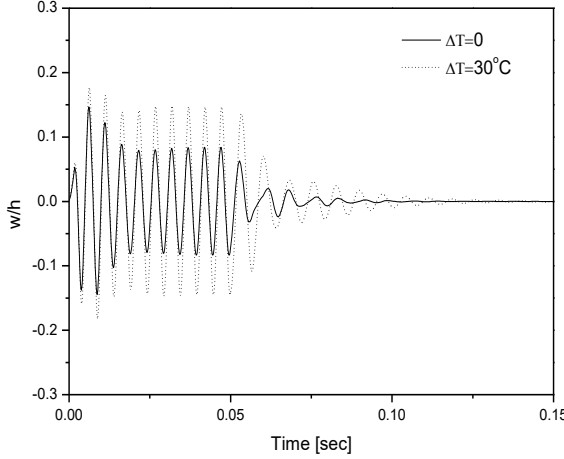

(**a**) Flutter suppression with thermal condition (Model (I), $\lambda = 1200$ )

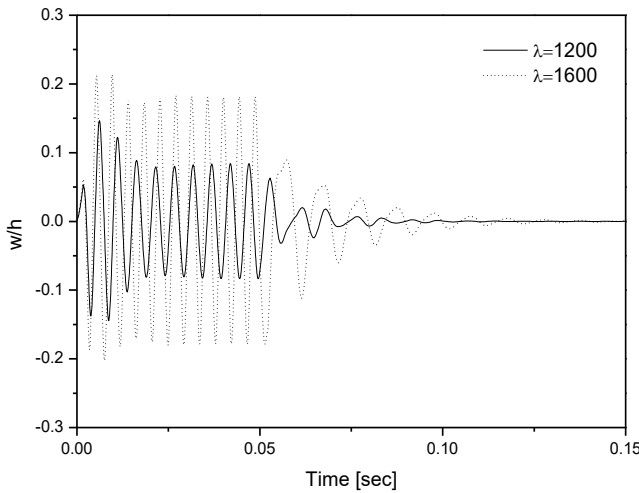

(**b**) Flutter suppression with aerodynamic pressure (Model (I), $\Delta T = 0$)

**Figure 10.** Flutter suppression behaviors under thermal and aerodynamic loads.

Finally, Figure 11 describes the flutter suppression motions of the structures based on piezoelectric patches. The control is also started after 0.05 s. In this case, the model (III) oscillated with high amplitude due to the flexibility of large antenna portion located at the center. Model (I) requires less suppression time compared to the other models. The results show that Model (I) is the most efficient for active flutter control because the piezoelectric materials for the Model (I) are the mostly widely displaced of the flutter suppression motions. It is very important to increase the portion of antenna components for the good radiofrequency radiation as an antenna structure. In addition, the patches are placed so as not to block the antenna parts, which means that the patches could be placed in a suboptimal position. It could degrade the performance of the sensor/actuator pairs. Ensuring the optimal conditions of the patches and the positions of the piezoelectric sensor and actuator are important for the antenna's performance.

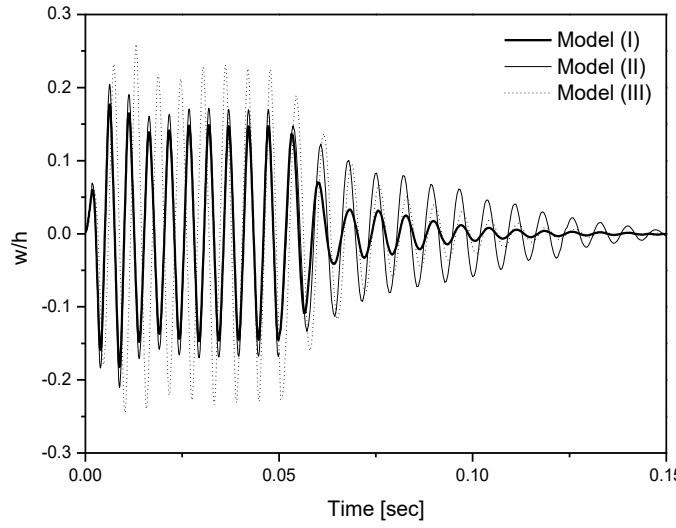

**Figure 11.** The flutter suppression of the designed models ($\lambda = 1200$, $\Delta T = 30$).

## 4. Conclusions

Active flutter control for smart-skin antenna structures are investigated using piezo-electric sensors and actuators. The antenna systems embedded in the airframe have many kinds of advantages. However, the integrated antenna models can be vibrated due to the aerodynamic loads. Then, the models will have severe deviations of the signal information. Therefore, the control of the smart-skin antenna models is very important from a structural point of view. For the active control, the linear quadratic regulator (LQR) control algorithm is applied. Prior to investigating the skin antenna, the honeycomb core model was analyzed with piezoelectric patches in various case studies on active vibration control conditions. The sizes as well as positions of the patches were studied in detail to measure the supersonic airflow. Accompanying these results, we selected the optimal positions of the piezoelectric patches for flutter control, and then skin antenna structures with thermal and aerodynamic loads were obtained for the effective control. The present results confirm that piezoelectric sensors and pairs of actuators pairs are placed 3/4 points from the air flow directions. Additionally, the use of patches of larger sizes and more outer layers from midplane are better for the flutter control. The LQR controller can effectively suppress the original flutter motions of the model with piezoelectric sensors and actuators.

**Author Contributions:** Conceptualization, C.-Y.L. and J.-H.K.; validation, C.-Y.L.; investigation, C.-Y.L.; writing—original draft preparation, C.-Y.L.; writing—review and editing, C.-Y.L. and J.-H.K.; project administration, C.-Y.L.; funding acquisition, C.-Y.L. All authors have read and agreed to the published version of the manuscript.

**Funding:** This research was supported by the Basic Science Research Program through the National Research Foundation of Korea (NRF) funded by the Ministry of Education (NRF-2019R1I1A3A01060180).

**Data Availability Statement:** The data are contained within the article.

**Conflicts of Interest:** The authors declare no conflict of interest.

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
