# Peer review of "Active Flutter Suppression of Smart-Skin Antenna Structures with Piezoelectric Sensors and Actuators"

_aerospace, doi:10.3390/aerospace8090257_

Round 1
Reviewer 1 Report
In this work active flutter control for smart-skin antenna structures are investigated using piezoelectric sensors and actuators.
The work is well organized and presents interesting results. Unfortunately, the author has already published this same study with the same palning, the same results, the same model and the same figures in a book (2015): "Active Flutter Control of Multifunctional Skin Antenna Structures considering Aerothermoelastic Characteristics", Student Number: 2009-21712, ed: d-collection. For this reason I have to reject the publication of this work in the journal.
Author Response
Responses to Reviewer 1’s Comments
Thanks for valuable suggestions to improve the quality of the manuscript.
Point 1: In this work active flutter control for smart-skin antenna structures are investigated using piezoelectric sensors and actuators.
The work is well organized and presents interesting results.
Unfortunately, the author has already published this same study with the same palning, the same results, the same model and the same figures in a book (2015): "Active Flutter Control of Multifunctional Skin Antenna Structures considering Aerothermoelastic Characteristics", Student Number: 2009-21712, ed: d-collection. For this reason I have to reject the publication of this work in the journal.
Response 1: Thank you for your advice. It seems that you pointed out the duplication problem with author’s doctoral thesis with student ID. However, there is no duplication problem between the author’s doctoral thesis and international journal papers. It is generally recommended and encouraged that the doctoral paper should be submitted to a high ranked journal paper for enhancing the quality of the paper through peer review process from experts. In this manuscript, all data have never been submitted to the other journal paper previously.
Thank you very much for your concern about ethical issues. All data has never been published in any journal, so there is no problem at all.
Thank you again.
If you have additional comments or questions for this manuscript, do not hesitate to inform authors as soon as possible.
Best regards,
Chang-Yull Lee

Reviewer 2 Report
The article's idea is an interesting one, but it is poorly proved for such a complex application. It shows some simulations for a very restricted situation and it was not mentioned anything about the restrictions or parameters which are not taken into consideration in this article for such an application. Actually, the author is presenting his hypothesis and the conclusion, but with almost no relevant demonstration. The active control is briefly mentioned taken into consideration that the flutter control is influenced by many factors which made such control very difficult to be realized. The author pretends that " the sizes as well as positions of patches are studied in detail for the supersonic airflow", but actually it is studied only on the very restricted approach. All his conclusions are not seriously sustained from a scientific or engineering point of view. Simulations prove some effect of the active control on the flutter effect on supersonic flow conditions, but not on the antennas electromagnetic parameters which are presented as the main application.
Reviewer 3 Report
This paper studies active flutter suppression using piezoelectric sensors and actuators. The topic is interesting, the paper is well organized and written. The following aspects should be improved before the paper can be suggested for publication.
1. An illustrative figure showing how the piezoelectric sensors and actuators are attached to the main structure is suggested.
2. It is suggested to add some explanations of the results from ref 7, 21, and 13. Are they from experiments or numerical simulations?
3. Figures 5, 6, 7. It is suggested to present the response without control measures.
4. Some discussions about the energy input required for the active control measures are suggested.
5. It suggested to clarity that the aerodynamic nonlinearity is not considered, which may remarkably affect the numerical results. Some references are given below.
Review of nonlinear panel flutter at supersonic and hypersonic speeds
Assessment of wind-induced nonlinear post-critical performance of bridge decks
Dowell, E., Edwards, J. and Strganac, T., 2003. Nonlinear aeroelasticity. Journal of aircraft, 40(5), pp.857-874.
6. Line 256, 3.2 should be 3.3.
Round 2
Reviewer 1 Report
The authors have modified the article according to the recommendations and I accept the authors' explanations about the possibility of plagiarism. I accept the article for publication in the journal in the present form
Reviewer 2 Report
The article suffered small changes but from a scientific or engineering point of view has nothing new to present as active control of a structure under aerodynamic generated flutter. Maybe only as an application for antennas.
But the article is well presented and could be accepted in this form.
Reviewer 3 Report
Comments are addressed properly. I suggest acceptance in the present form.